# Experiencing a significant life event during the COVID-19 pandemic: The role of perceived control

**Sonja Radjenovic** *, **Christina Ristl, Jana Nikitin**

Department of Developmental and Educational Psychology, University of Vienna, Vienna, Austria

* radjenovic.sonja@yahoo.com

## Abstract

The COVID-19 pandemic represents a context that might influence how people experience significant life events (such as marriage, job change, or relocation). In the present research, we argue that one important factor of how positively or negatively the pandemic influences the experience of significant life events is how much control people perceive in the situation. An online sample of $N = 882$ participants aged 18–82 years reported a significant life event that occurred during the COVID-19 pandemic. The participants described whether the pandemic influenced the life event, to what extent they perceived control over the life event and how they experienced the life event in terms of valence (i.e., positively or negatively). The results showed that the self-reported pandemic influence was significantly associated with the life-event experience. Furthermore, perceived control partly mediated this association. The results persisted even after controlling for the age of the participants. The present research underscores the important role of control for experiencing significant life events in challenging times such as during the pandemic.

## Introduction

Significant life events have a clear time frame, interfere with everyday life, and are perceived as personally important [1]. They include life changes such as marriage, separation, relocation, illness, loss of a loved one, etc. [2]. Although not all life events are equally frequent across adulthood (e.g., marriage is more frequent in younger adulthood) [3], many life events can be experienced at any age. Significant life events can differ in their emotional experience. The experience of significant life events is often operationalized in terms of valence (e.g., positive or negative, desirable or undesirable, enriching or distressful) [4]. How did people experience significant life events during the COVID-19 pandemic? COVID-19 is a respiratory disease [5, 6] that led many countries to order lockdowns that reduced social contacts. This resulted in partly drastic changes in people's lives [7, 8].

There are reasons to believe that the pandemic influenced the experience of significant life events. The pandemic might lead people to appraise otherwise harmless or neutral events as threatening, because ongoing worry consumes individuals' coping resources [9]. This can lead

**Data Availability Statement:** All relevant data are within the paper and its Supporting Information files.

**Funding:** This research was supported by Grant 100019_159399 from the Swiss National Science Foundation (https://www.snf.ch/en) (PI: Jana

Nikitin) and by Grant STWF-16-019 from the UZH Foundation (https://www.uzhfoundation.ch/) (PI: Jana Nikitin). The funders had no role in study design, data collection and analysis, decision to publish, or preparation of the manuscript.

**Competing interests:** The authors have declared that no competing interests exist.

to more negative emotional responses to significant life events experienced during the pandemic. On the other hand, the perceived influence of the pandemic does not necessarily have to be negative. The pandemic allowed some people to withdraw from the stressful everyday life, get off the "hamster wheel" and concentrate on subjectively important aspects of life [10, 11]. This can lead to more positive emotional responses to significant life events because more time and resources can be put into dealing with them. Accordingly, some studies found that people were under considerable physical and psychological well-being risk during the pandemic [12], whereas other studies found similar resilience and life satisfaction during the pandemic as prior to it [13, 14]. Obviously, not all people experience the pandemic in the same way.

We argue that an important factor that might explain the experience of life events during the pandemic is perceived control. Perceived control, defined as individual's perceived capacity to influence a certain incident, is fundamental for the mobilization of resources and coping strategies in challenging situations [15], leading to better well-being and psychological outcomes [16, 17]. The changes associated with the pandemic might make it either more difficult or easier for people to keep control over their life events. The pandemic can be either seen as an additional stress factor alongside a significant life event [7], with little influence of individual's behavior on the outcome of the situation. Or the pandemic might make it easier to maintain control over life events, because it allows people to step back from the stressful daily life and focus on the most important aspects of life [10, 11], providing more coping resources and perceived control over life events. Studies [18, 19] found that perceived control may act as a protective factor during the pandemic. Perceived control was found to buffer the effects of pandemic severity on distancing oneself from the pandemic situation in terms of psychological distance [20], and to mitigate the negative impact of the pandemic on mental health and well-being [19, 21–24]. Further research found a positive association between perceived control and specific positive outcomes during the pandemic, such as participation in social life [25], preventive health behaviors [26], and subjective health competence [23, 27]. Thus, the pandemic does not automatically have to trigger negative experiences; it is the sense of control that determines how people respond to it [28]. Based on this evidence, we argue that sense of control is the exploratory mechanism for the impact of the pandemic on the experience of significant life events.

In addition, we explore possible age effects in the experience of the life event during the pandemic. There are heterogeneous findings on age differences in subjective well-being during the pandemic. According to some studies, older adults worried relatively less, reported higher well-being, less loneliness and depressive symptoms, more positive events, better coping ability, and less affective reactivity to pandemic stressors [7, 29]. However, other studies found that the pandemic had a negative effect on older adults' mental health and well-being [30]. Finally, some studies found no age-related associations between the pandemic and psychological outcomes [31, 32]. Obviously, some older adults are affected by the pandemic, whereas others are not [31, 33], so it is unclear whether age acts as a risk or as a protective factor in this context. Thus, we would like to shift the focus away from age differences to people's life circumstances. We argue that people who are in control of the situation would experience the life event more positively, irrespective of age. Perceived control is an important factor throughout people's lives, though it shows some development changes over the course of life [34]. Specifically, young adults reach control by changing the environment to fit their goals (i.e., assimilative strategy), whereas with increasing age, people rather adjust the aspiration level of their goals to compensate for experienced losses (i.e., accommodative strategy) [35]. Despite these different coping strategies, all people strive for high levels of perceived control in their lives

[34]. We argue therefore, that if the pandemic reduces people's perceived control over their life events, they experience lower valence of the life events, irrespective of age.

The aim of the study is to investigate whether the pandemic affects the way we experience significant life events and to explore the role of age and perceived control over the life event in this effect. We hypothesize that self-reported positive influence of the pandemic on the life event has a positive association with the life-event experience (H1a), and that self-reported negative influence of the pandemic on the life event has a negative association with the life-event experience (H1b). We further hypothesize that perceived control over the life event is the exploratory mechanism of these associations. Accordingly, perceived control positively predicts the life-event experience (H2) and is in turn positively predicted by the self-reported positive influence of the pandemic on the life event (H3a) and negatively predicted by the self-reported negative influence of the pandemic on the life event (H3b). We expect that these associations persist after controlling for participants' age.

## Methods

### Participants and procedure

We tested our hypotheses in a large, age-heterogenous sample with participants from German-speaking countries. The ethical committee of the University of Basel approved the collection of data for this study. The participants were recruited via Respondi, an ISO-certified recruitment company that guarantees the recruitment of highly motivated participants through fair incentives and personal support as well as regular checks on the identity, plausibility, and response behavior of participants. Participants are recruited via online campaigns (mostly Facebook or other social media platforms).

At the beginning of the study, participants provided informed consent and were guaranteed that all data collected would be kept confidential. Data was anonymized and it was not possible to link the data to individual participants. Participants were reimbursed for completing the questionnaire of approximately 20–30 minutes length with 3.60 €. Data was collected online between June 23 and July 8, 2020.

Only participants who reported at least one significant life event in the past two years (selected from a list in a screening questionnaire) were invited to the study, resulting in $N = 6{,}688$ participants. At the beginning of the questionnaire, participants were asked to select one significant life event (from the same list as in the screening questionnaire) that had the greatest impact on their lives and answered follow-up questions regarding this life event. For the present study, we included only participants who reported a significant life event that occurred since the declaration of the pandemic in March 2020, resulting in a sample of $N = 882$ participants.

### Measures

**Significant life events.** Adapted from HILDA (Household, Income, and Labour Dynamics in Australia) [36] and SHP (Swiss Household Panel) [37], participants read a list of 19 life events such as relocation, unemployment, or serious illness/injury (see S1 File for all items and their frequency in the sample) and chose an event they experienced in the last two years that had the greatest impact on their lives. If none of the listed events was applicable, the participants selected a category "other" and described their specific life event. For the individual analyses, events with similar contents were clustered. The resulting seven (clusters of) events are (1) "relocation" (including relocation to another country, city or place of residence), (2) "birth of a child", (3) "own serious illness or injury", (4) "serious illness or injury of a loved one" (including serious illness or injury of a partner, a close family member or a friend), (5) "loss of

a loved one" (including loss of a partner, family member or a friend), (6) "job change", and (7) "unemployment".

**Life-event experience.** Participants rated the significant life event using the items "positive", "negative", "desirable", "undesirable", distressing", and "enriching" (1 = *not at all* to 7 = *very*). Negatively worded items were recoded, and all items were aggregated to one scale (Cronbach's α = .94). Higher scores indicate more positive and less negative experience. The items for life-event experience were inspired by the valence scale of Luhmann and colleagues [1], with slightly modified wording.

**Perceived control over the life event.** Participants specified to what extent they had control over the onset, the process, and the outcome of the significant life event (1 = *no control at all* to 7 = *full control*). These three items were aggregated to a scale indicating overall perceived control. The scale was inspired by the Perceived Control Over Stressful Events Scale which includes past, present, and future control items [38]. The self-developed scale had a Cronbach's α = .91.

**Self-reported influence of the pandemic on the life event.** Participants were asked to what extent the pandemic influenced (a) the life event in terms of influence intensity, positive influence, and negative influence (1 = *not at all* to 7 = *very much*) and (b) the experience of the life event in terms of influence intensity, positive influence, and negative influence (1 = *not at all* to 7 = *very much*). The six items were aggregated in three scales describing the overall influence intensity, positive influence, and negative influence. Overall influence intensity was used as a continuous control variable. The self-developed scales had a Cronbach's α = .85 for influence intensity, α = .85 for positive influence, and α = .87 for negative influence.

## Data-analytical strategy

Hierarchical regression analyses were used to test the hypotheses. Age as a continuous variable was included in a second step to test its incremental contribution to the explained variance. The analyses were conducted both across the overall sample and for the seven (clusters of) individual life events (using the IBM SPSS 25 software [39]), to rule out that the results from the overall sample are confounded by the content of specific life events. We expected similar results within individual life events as in the overall sample, as perceived control should be equally important for the experience of the life events, irrespective of their specific content. Finally, to rule out that the results are confounded by differences in the relevance of the pandemic for different life events, we controlled for perceived intensity of the influence of the pandemic on the life events in the analyses. In addition, gender and education were included as control variables. Analyses without gender and education as control variables can be found in the S3 and S4 Tables in S1 File.

Although the correlational nature of the study does not allow causal explanations, we explored whether perceived control could be considered as a mediator of the association between the positive and negative influence of the pandemic on the life event and the experience of the life event. Mediation analyses were conducted in the overall sample and across the seven (clusters of) individual life events, including self-reported influence intensity of the pandemic on the life event, gender, and education as covariates.

To ensure sufficient statistical power for the analyses within the (clusters of) individual life events, we conducted a power analysis. A large meta-analysis on subjective well-being and significant life events showed medium to large effect sizes in previous research [40]. Thus, we expected medium to large effect sizes in our analyses. We used the software program G*Power [41] to conduct a power analysis for the number of participants needed for each life event. With .05 alpha error probability, an effect size $f^2$ of .20, a power of .80, and

two predictors, 52 participants were at least needed for a single life event. All life events met (or nearly met) this requirement (see Table 2). The life events that did not meet this requirement were not considered for individual analyses. Cohen's indices were used to indicate the goodness of fit of the regression models, where $R^2 < 0.02$ represents very low goodness of fit, $R^2 = 0.02$ to $0.12$ low goodness of fit, $R^2 = 0.13$ to $0.26$ moderate goodness of fit, and $R^2 > 0.26$ high goodness of fit [42].

## Results

### Descriptive analyses

The age and gender distribution did not significantly differ between the overall sample of 6'688 participants and the present study sample of 882 participants ($ps \geq .11$; 52.6% male, 47.2% female, 0.2% neither nor; age range 18–82 years). There was a similar distribution of participants across age decades (144 to 172 per decade), apart from participants above 70, who accounted for 89 participants. Participants were from Germany (78.1%), Austria (13.7%), Switzerland (7.8%), and some other countries (0.3%). They graduated from a university/college (28.5%), a higher vocational-school (15%), a high-school (17.1%), a vocational training (31.9%), an obligatory school (4.2%), and 3.4% specified some other degree. The sample consists of up to 10% more individuals with a university/college degree than in the general population of German-speaking countries, up to 10% fewer individuals with a high school diploma, and with vocation school percentages being representative for the general population [43]. Employment status (60%) is smaller [44], and the marital status almost the same as in the general population [45].

Correlation analyses showed a significant negative correlation between age and all measured variables, except for a non-significant correlation with self-reported negative influence of the pandemic on the life event. Perceived control correlated significantly with all variables except for the self-reported influence intensity on the life event: Perceived control was positively associated with the life event experience and the self-reported positive influence of the pandemic on the life event, and negatively associated with the self-reported negative influence of the pandemic on the life event. Life event experience was further significantly negatively associated with the self-reported negative influence and influence intensity, and positively correlated with the self-reported positive influence of the pandemic. Finally, influence intensity was positively associated with both the positive and negative influence of the pandemic. For more information on descriptive statistics and bivariate correlations see Table 1. For the descriptive statistics of (clusters of) individual life events see Table 2, and for all 19 individual life events see S2 Table in S1 File.

**Table 1. Means, standard deviations and correlations.**

| Variable | n | M | SD | α | 1. | 2. | 3. | 4. | 5. |
|---|---|---|---|---|---|---|---|---|---|
| 1. Age | 882 | 47.23 | 16.00 | - | | | | | |
| 2. Perceived control | 882 | 3.32 | 2.10 | .91 | -.28** | | | | |
| 3. Life event experience | 882 | 3.47 | 2.31 | .94 | -.27** | .69** | | | |
| 4. Influence intensity | 882 | 3.99 | 2.25 | .85 | -.09* | -.05 | -.09* | | |
| 5. Positive influence | 882 | 2.61 | 1.70 | .85 | -.22** | .40** | .39** | .12** | |
| 6. Negative influence | 882 | 3.90 | 2.11 | .87 | -.05 | -.16** | -.24** | .80** | -.03 |

*Note*. Influence intensity, and positive and negative influence correspond to the self-reported influence of the pandemic on the life event. *$p < .05$.

**$p < .01$.

**Table 2. Descriptive statistics for (clusters of) individual life events.**

| Life event | n | Age | | Positive influence | | Negative influence | | Life-event experience | | Perceived control | |
|---|---|---|---|---|---|---|---|---|---|---|---|
| | | M | SD | M | SD | M | SD | M | SD | M | SD |
| Relocation | 107 | 38.43 | 15.95 | 3.09 | 1.63 | 3.75 | 1.85 | 5.25 | 1.26 | 5.46 | 1.12 |
| Birth of a child | 49 | 38.68 | 10.19 | 3.26 | 1.83 | 4.11 | 1.79 | 6.24 | 0.72 | 4.10 | 1.91 |
| Own serious illness or injury | 67 | 56.79 | 15.29 | 2.46 | 1.59 | 3.69 | 2.23 | 1.73 | 1.14 | 2.75 | 1.49 |
| Serious illness or injury of a loved one | 112 | 54.71 | 13.95 | 1.95 | 1.19 | 4.42 | 2.07 | 1.53 | 0.89 | 1.91 | 1.29 |
| Loss of a loved one | 184 | 54.05 | 14.56 | 1.87 | 1.35 | 3.65 | 2.33 | 1.57 | 0.84 | 1.66 | 1.32 |
| Job change | 73 | 39.12 | 10.45 | 3.49 | 1.85 | 3.15 | 1.83 | 5.43 | 1.54 | 4.77 | 1.55 |
| Unemployment | 75 | 44.71 | 13.27 | 2.27 | 1.66 | 5.48 | 1.81 | 1.98 | 1.41 | 2.00 | 1.43 |

*Note*. Life events that met the criteria of the power analysis. Life events' names were abbreviated: relocation ("relocation to another country, to another place of residence"), serious illness or injury of a loved one ("serious illness or injury of the partner, a close family member or a friend"), loss of a loved one ("loss of the partner, a close family member or a friend"). *n*, *M* and *SD* are used to describe subsample number, mean and standard deviation respectively.

## H1: Self-reported influence of the pandemic on the life event and life-event experience

Across all life events, we found a significant positive association between the self-reported positive influence of the pandemic on the life event and life-event experience (β = .36, *p* < .001, 95% CI [0.41, 0.57]) (see Fig 1A). The positive influence of the pandemic explained 15.7% of variance in life-event experience, indicating moderate goodness of fit. Age explained 3.7% additional variance (β = -.20, *p* < .001, 95% CI [-0.04, -0.02]; low goodness of fit). In addition, there was a significant negative association between the self-reported negative influence of the pandemic on the life event and life-event experience (β = -.47, *p* < .001, 95% CI [-0.62, -0.40]) (see Fig 1B), with 8.2% explained variance (low to moderate goodness of fit). Age explained 7% additional variance in this model (β = -.27, *p* < .001, 95% CI [-0.05, -0.03]; low to moderate goodness of fit).

The results of the analyses for the individual life events showed a significant relationship between the self-reported positive influence of the pandemic on the life event and life-event experience for "serious illness or injury of a loved one", "loss of a loved one", "job change" and "unemployment". For the self-reported negative influence of the pandemic on the life event and life-event experience, there were significant associations for "birth of a child", "loss of a loved one", "job change" and "unemployment". Age did not contribute significantly to the variance explained. For results see Model 1 (positive influence) and 2 (negative influence) in Table 3.

## H2: Perceived control and life-event experience

Across all life events, perceived control significantly positively predicted the experience of the life event (β = .66, *p* < .001, 95% CI [0.68, 0.78]), accounting for 46.8% of the explained variance in the life-event experience (high goodness of fit; see Fig 1C). Age explained 0.7% additional variance in the life-event experience (β = -.09, *p* = .001, 95% CI [-0.02, -0.01]; low goodness of fit; see Fig 1D). The relationship between perceived control and the life-event experience was significant for all (clusters of) individual life events (see Model 3 in Table 3). The only life event that showed a significant relationship between age and the life-event experience was "birth of a child" (see Model 3 in Table 3).

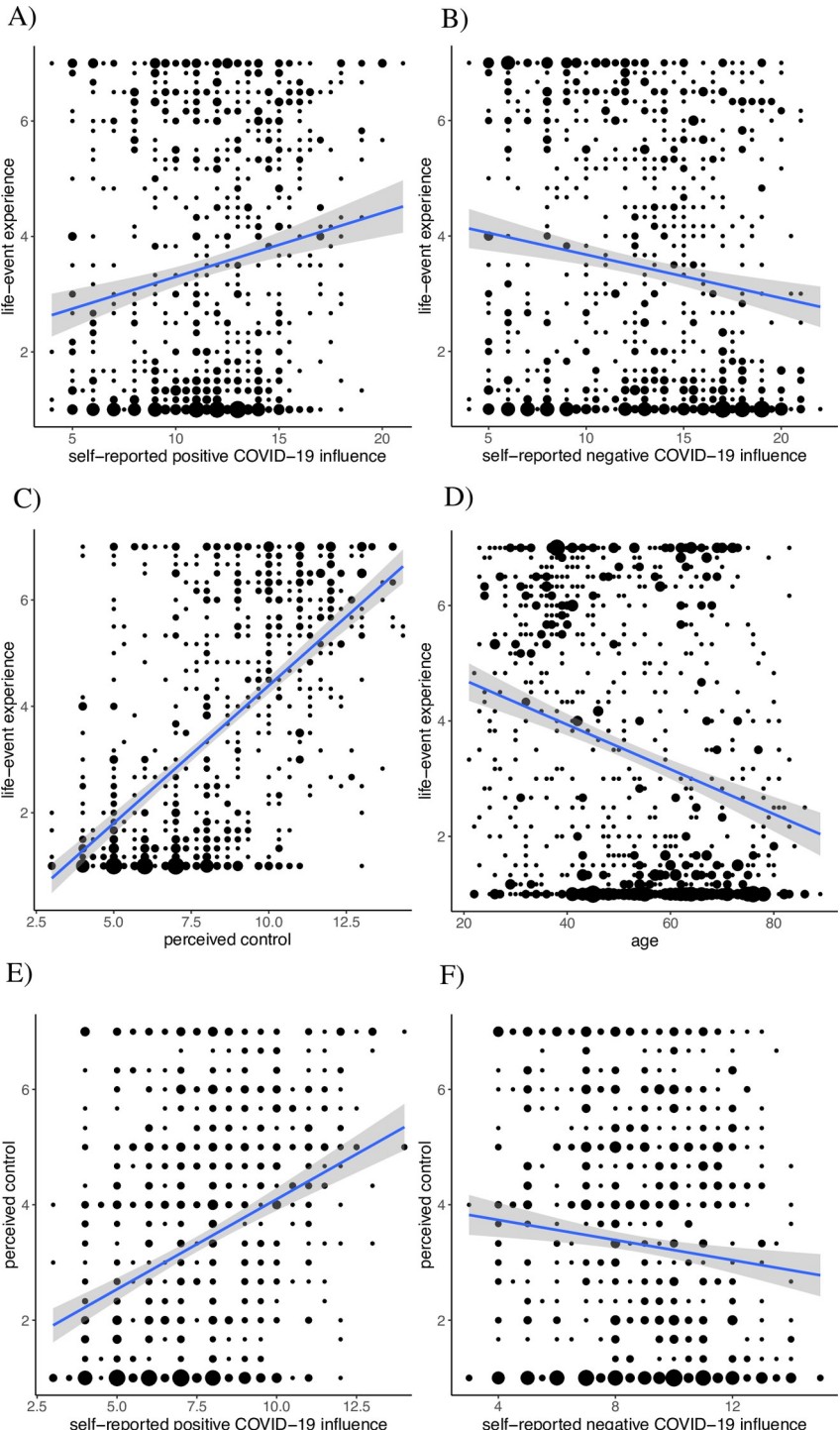

**Fig 1. Regression lines with corresponding confidence intervals.** (A, B) Self-reported COVID-19 influence and life-event experience, (C) Perceived control and life-event experience, (D) Age and life-event experience, (E, F) Self-Reported COVID-19 influence and perceived control. *Note*. The size of the bubbles represents the number of participants. Controlled for self-reported COVID-19 influence intensity, gender, and education.

**Table 3. Results for individual life events with life-event experience as the outcome variable.**

| | | β | p | 95% CI LL | UL | R2 | ΔR² |
|---|---|---|---|---|---|---|---|
| Relocation | | | | | | | |
| Model 1 | | | | | | .016 | .001 |
| | Positive influence | -.13 | .19 | -.257 | .050 | | |
| | Age | -.04 | .71 | -.018 | .013 | | |
| Model 2 | | | | | | .024 | .004 |
| | Negative influence | -.30 | .09 | -.432 | .030 | | |
| | Age | -.07 | .50 | -.021 | .010 | | |
| Model 3 | | | | | | .133 | .002 |
| | Perceived control | **.35** | **< .001** | **.187** | **.610** | | |
| | Age | -.05 | .62 | -.019 | .011 | | |
| Birth of a child | | | | | | | |
| Model 1 | | | | | | 002 | .053 |
| | Positive influence | .10 | .55 | -.091 | .169 | | |
| | Age | .24 | .13 | -.005 | .039 | | |
| Model 2 | | | | | | .095 | .040 |
| | Negative influence | **-.36** | **.04** | **-.277** | **-.010** | | |
| | Age | .21 | .16 | -.004 | .035 | | |
| Model 3 | | | | | | .234 | .174 |
| | Perceived contro | **.53** | **.003** | **.071** | **.326** | | |
| | Age | **.52** | **.003** | **.013** | **.061** | | |
| Own serious illness or injury | | | | | | | |
| Model 1 | | | | | | .004 | .030 |
| | Positive influence | .07 | .57 | -.124 | .225 | | |
| | Age | .17 | .16 | -.005 | .031 | | |
| Model 2 | | | | | | .001 | .028 |
| | Negative influence | -.06 | .77 | -.248 | .184 | | |
| | Age | .17 | .17 | -.006 | .031 | | |
| Model 3 | | | | | | .130 | .022 |
| | Perceived control | **.26** | **.04** | **.012** | **.385** | | |
| | Age | .15 | .21 | -.007 | .029 | | |
| Serious illness or injury of a loved one | | | | | | | |
| Model 1 | | | | | | .068 | .006 |
| | Positive influence | **.25** | **.01** | **.041** | **.325** | | |
| | Age | -.08 | .42 | -.017 | .007 | | |
| Model 2 | | | | | | .025 | .020 |
| | Negative influence | -.32 | .08 | -.288 | .015 | | |
| | Age | -.14 | .14 | -.021 | .003 | | |
| Model 3 | | | | | | .181 | .016 |
| | Perceived control | **.39** | **< .001** | **.150** | **.390** | | |
| | Age | -.13 | .15 | -.019 | .003 | | |
| Loss of a loved one | | | | | | | |
| Model 1 | | | | | | .069 | .014 |
| | Positive influence | **.28** | **< .001** | **.087** | **.263** | | |
| | Age | .12 | .09 | -.001 | .015 | | |
| Model 2 | | | | | | .050 | .012 |
| | Negative influence | **-.47** | **.001** | **-.272** | **-.066** | | |
| | Age | .11 | .12 | -.002 | .014 | | |

(*Continued*)

**Table 3.** (Continued)

| | | β | p | 95% CI LL | UL | R2 | ΔR² |
|---|---|---|---|---|---|---|---|
| Model 3 | | | | | | .140 | .011 |
| | Perceived control | **.25** | **< .001** | **.073** | **.250** | | |
| | Age | .11 | .14 | -.002 | .014 | | |
| Job change | | | | | | | |
| Model 1 | | | | | | .067 | .011 |
| | Positive influence | **.27** | **.03** | **.026** | **.420** | | |
| | Age | .11 | .35 | -.018 | .052 | | |
| Model 2 | | | | | | .186 | .032 |
| | Negative influence | **-.59** | **< .001** | **-.728** | **-.273** | | |
| | Age | .19 | .09 | -.004 | .061 | | |
| Model 3 | | | | | | .353 | .003 |
| | Perceived control | **.54** | **< .001** | **.332** | **.732** | | |
| | Age | .06 | .55 | -.021 | .040 | | |
| Unemployment | | | | | | | |
| Model 1 | | | | | | .181 | .002 |
| | Positive influence | **.44** | **< .001** | **.195** | **.557** | | |
| | Age | .05 | .66 | -.018 | .028 | | |
| Model 2 | | | | | | .096 | .000 |
| | Negative influence | **-.67** | **.006** | **-.890** | **-.155** | | |
| | Age | .003 | .98 | -.024 | .024 | | |
| Model 3 | | | | | | .257 | .000 |
| | Perceived control | **.52** | **< .001** | **.294** | **.726** | | |
| | Age | -.02 | .87 | -.024 | .021 | | |

*Note.* Controlled for influence intensity, gender, and education. CI = confidence intervals. Significant results ($p < .05$) are presented in bold.

## H3: Self-reported influence of the pandemic on the life event and perceived control

Across all life events, the self-reported positive influence of the pandemic on the life event was positively associated with perceived control (β = .37, $p < .001$, 95% CI [0.38, 0.53]; see Fig 1E), accounting for 16.1% of the explained variance in perceived control (moderate goodness of fit). Age explained 3.7% additional variance in perceived control (β = -.20, $p < .001$, 95% CI [-0.03, -0.02]; low goodness of fit). The self-reported negative influence of the pandemic on the life event was negatively associated with perceived control (β = -.33, $p < .001$, 95% CI [-0.43, -0.22]; see Fig 1F), accounting for 4.1% of the explained variance in perceived control (low goodness of fit). Age explained 7.2% additional variance (β = -.27, $p < .001$, 95% CI [-0.04, -0.03]; low to moderate goodness of fit). For the (clusters of) individual life events, significant associations were found between the self-reported positive influence of the pandemic on the life event and perceived control for "loss of a loved one", "job change", "unemployment" and "birth of a child", the latter of which included a significant age effect. Regarding the negative influence of the pandemic on the life event and perceived control, there was a significant association for "birth of a child", which also included a significant age effect. For detailed results see Model 1 and 2 in Table 4.

## Exploratory analyses

The mediation analysis revealed a significant partial mediation effect of perceived control on the association between self-reported positive influence of the pandemic on the life event and

**Table 4. Results for individual life events with perceived control as the outcome variable.**

| | | β | p | 95% CI | | R2 | ΔR² |
|---|---|---|---|---|---|---|---|
| | | | | LL | UL | | |
| Relocation | | | | | | | |
| Model 1 | | | | | | .000 | .005 |
| | Positive influence | .01 | .96 | -.135 | .142 | | |
| | Age | .07 | .47 | -.009 | .019 | | |
| Model 2 | | | | | | .001 | .005 |
| | Negative influence | -.02 | .91 | -.222 | .197 | | |
| | Age | .07 | .49 | -.009 | .019 | | |
| Birth of a child | | | | | | | |
| Model 1 | | | | | | .183 | .239 |
| | Positive influence | **.38** | **.003** | **.138** | **.646** | | |
| | Age | **-.51** | **< .001** | **-.138** | **-.053** | | |
| Model 2 | | | | | | .052 | .324 |
| | Negative influence | **-.32** | **.03** | **-.640** | **-.043** | | |
| | Age | **-.58** | **< .001** | **-.154** | **-.066** | | |
| Own serious illness or injury | | . | . | | | . | . |
| Model 1 | | | | | | 003 | 010 |
| | Positive influence | 06 | 65 | -.179 | .286 | | |
| | Age | .10 | .42 | -.014 | .034 | | |
| Model 2 | | | | | | .020 | .009 |
| | Negative influence | -.24 | .26 | -.447 | .122 | | |
| | Age | .10 | .44 | -.015 | .033 | | |
| Serious illness or injury of a loved one | | | | | | . | . |
| Model 1 | | | | | | .019 | .001 |
| | Positive influence | .14 | .15 | -.057 | .367 | | |
| | Age | .03 | .79 | -.016 | .021 | | |
| Model 2 | | | | | | .006 | .000 |
| | Negative influence | -.14 | .43 | -.314 | .135 | | |
| | Age | -.01 | .92 | -.019 | .017 | | |
| Loss of a loved one | | | | | | | |
| Mode 1 | | | | | | .029 | .000 |
| | Positive influence | **.18** | **.02** | **.027** | **.318** | | |
| | Age | .02 | .78 | -.011 | .015 | | |
| Mode 2 | | | | | | .007 | .000 |
| | Negative influence | -.18 | .25 | -.270 | .070 | | |
| | Age | .01 | .87 | -.012 | .014 | | |
| Job change | | | | | | | |
| Model 1 | | | | | | .078 | .010 |
| | Positive influence | **.42** | **< .001** | **.042** | **.441** | | |
| | Age | .11 | .39 | -.020 | .051 | | |
| Model 2 | | | | | | .015 | .017 |
| | Negative influence | -.18 | .24 | -.415 | .105 | | |
| | Age | .14 | .27 | -.016 | .058 | | |
| Unemployment | | | | | | | |
| Model 1 | | | | | | .170 | .000 |
| | Positive influence | **.42** | **< .001** | **.193** | **.537** | | |
| | Age | .01 | .90 | -.020 | .023 | | |

*(Continued)*

**Table 4.** (Continued)

| | | β | p | 95% CI | | R2 | ΔR² |
| --- | --- | --- | --- | --- | --- | --- | --- |
| | | | | LL | UL | | |
| Model 2 | | | | | | .039 | .002 |
| | Negative influence | -.41 | .08 | -.688 | .037 | | |
| | Age | -.05 | .68 | -.029 | .019 | | |

*Note.* Controlled for influence intensity, gender, and education. CI = confidence intervals. Significant results are in bold.

the life-event experience (direct path: $B = .20$, $p < .001$, 95% CI [0.13, 0.27]; indirect path: $B = .69$, $p < .001$, 95% CI [0.63, 0.74]; see Fig 2A), as well as for the self-reported negative influence of the pandemic on the life event and the life-event experience (direct path: $B = -.28$, $p < .001$, 95% CI [-0.37–0.20]; indirect path: $B = .72$, $p < .001$, 95% CI [0.66, 0.77]; see Fig 2B).

Regarding (clusters of) individual significant life events, there were significant partial mediation effects in 3 out of 7 models for positive pandemic influence and none for negative pandemic influence (see S5 Table in S1 File).

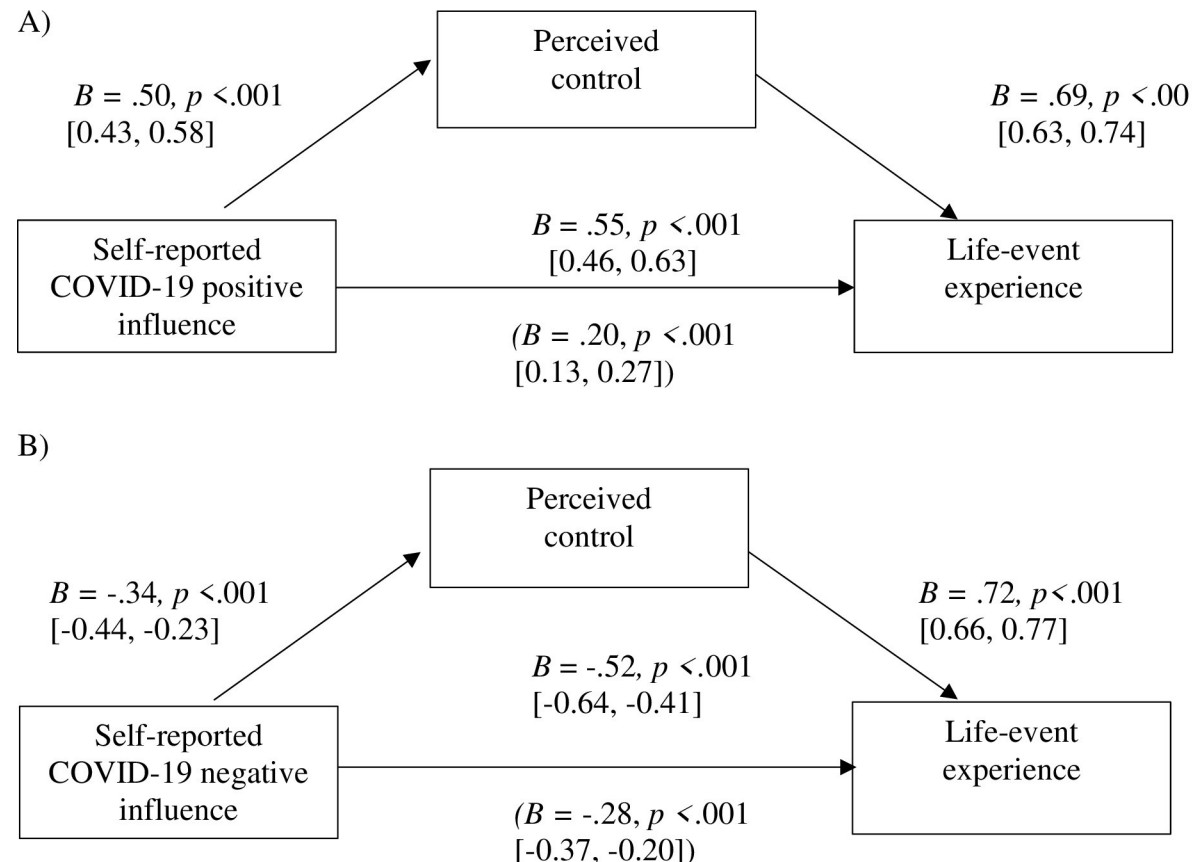

**Fig 2. Mediation analyses for perceived control as a mediator between self-reported influence of the pandemic on the life event and life-event experience in the overall sample.** *Note.* Analyses were conducted using Hayes Model 4 with 5,000 bootstraps. Total effects and direct effects (in parentheses) are depicted. Self-reported influence intensity of the pandemic on the life event, gender, and education were considered covariates.

## Discussion

The COVID-19 pandemic is a challenging time, not only due to its life-threatening health consequences, but also due to changes in people's daily lives. The findings of the present study demonstrate that the pandemic might affect how we experience significant life events (H1). An important factor in this context is the feeling of having control over the life event. Our results show that a substantial portion (almost a half) of the variance in the life-event experience was explained by perceived control over the event (H2). Perceived control was not only associated with a more positive life-event experience, but also partially explained the relationship between self-reported positive and negative influence of the pandemic on the event and experience of the event (results of exploratory analyses). A mediation analysis of correlational data does not prove a mediation effect (for a detailed discussion see [46]) and should be interpreted with utmost caution. Nevertheless, our findings are consistent with previous research that has identified perceived control as an important factor in well-being and coping with difficult times [18, 38], and should be followed up in future research. The self-reported positive pandemic influence was associated with more perceived control, and the self-reported negative pandemic influence was associated with less perceived control (H3).

Although age was only weakly associated with the experience of the life event, the associations were statistically significant, suggesting that age explained the experience of the life event in addition to the self-reported influence of the pandemic and perceived control over the life event. In general, life events were experienced less positively by older participants, and older age was associated with lower perceived control (results of H3). Both findings are consistent with previous evidence that older adults experience fewer developmental gains and more developmental losses [47] as well as less positive life events [48].

In terms of individual life events, the results of our study suggest some positive influence of the pandemic on the life events "serious illness or injury of a loved one", "loss of a loved one", "job change," and "unemployment". We can only speculate why this is the case. What these life events have in common is that they tend to be challenging. In these challenging times, the pandemic might give at least some people more time for themselves and for solving their problems, as well as more space for grieving and overcoming challenges; perhaps also less pressure to solve certain problems (such as unemployment) quickly. However, there was also some negative influence of the pandemic on the life events "birth of a child", "loss of a loved one", "job change" and "unemployment." This could be due to the direct influence of the pandemic on these life events (e.g., not being able to visit a loved one after giving birth, being exposed through hospitalization, losing a job or a loved one as a consequence of COVID-19) or indirectly (e.g., having more time to ruminate or being less able to actively address the issue). In fact, the results showed that across all individual life events, perceived control was an influential contributor to the explained variance in the experience of life events, partially mediating the experience of the event in about one-fifth of individual life events. For self-reported positive pandemic influence, this was the case for "loss of a loved one", "job change", "unemployment" and "birth of a child". Obviously, control is particularly important to the positive experience of these events. For self-reported negative pandemic influence, this was the case for "birth of a child.". Thus, the birth of a child seems to be particularly closely associated with the need for control [49]. Older age was associated with less perceived control over the "birth of a child" but with more positive experience. This could be due to anticipating a child longer with increasing age, or to the experience of a family member and not necessarily one's own childbirth (e.g., becoming a grandparent).

## Limitations and implications for future research

Although the study is unique because of the wide age range and variety of different life events, it also has some limitations. One limitation is the retrospective assessment of a life event, which could lead to recall biases in participants' responses. Another limitation is that the design of our study does not allow comparisons between life events before and during the pandemic. Although this limitation is mitigated by the subjectively reported influence of the pandemic on the life event, longitudinal data with pre and post pandemic assessment would be most optimal, as would comparisons of individuals who experienced significant life events with those who did not. In addition, while the approach of recruiting participants online has clear benefits because it enables rapid data collection of large samples, it also introduces a significant selection bias, especially when it comes to older adults. Due to this bias, older adults who regularly use the internet are overrepresented in the study population.

Despite the many analyses used in this study, we did not apply alpha-level corrections because the use of these procedures reduces the power and increases the type II error [50]. In addition, they place too much emphasis on significance levels rather than effect sizes and explained variance, which we have reported here, and which are better indicators of meaningful results [51].

Given the impact of the pandemic on mental health [19], data on depression or anxiety could have added to our knowledge of the life-event experience. Mental health issues might impact perception, including perceived control over the life events, the valence of the life events' experience, and the pandemic itself, and therefore, when considered, could lead to different results.

Finally, it is open whether the results would vary by geographic location due to different social and health care systems or different handling of the pandemic situation by the government. Regarding the national response of German-speaking countries (Germany, Austria, and Switzerland) to the pandemic, it should be noted that Austria and Switzerland implemented the regulations in a largely uniform manner in the various federal states and cantons, whereas in Germany different policies were pursued depending on the federal state [52]. Austria imposed a first lockdown early on, Germany and Switzerland followed a few days later. However, there was never a complete lockdown in Germany, meaning that contact with other people was allowed if following social distancing [53]. In late April 2020, Austria was the Germany-speaking country with the most random COVID-19 testing, while Germany and Switzerland had tested only vulnerable populations and individuals with symptoms [52]. The Swiss government did not introduce a mask policy at the beginning of the pandemic, appealing to personal responsibility [54]. Given this large heterogeneity in the regulations, we argue that the personal influence experienced by individuals may be more important than location. However, if we could identify which regulations were most influential for which life events, it would improve our understanding of the role of the pandemic in the experience of life events.

The question is not only whether the results and mechanisms might be different across the assessed countries, but also in other cultures. On the one hand, researchers suggest that primary control is fundamental to every individual, regardless of their cultural background [55]. On the other hand, it is known that individuals from collectivistic cultures are more oriented towards external sources of support, such as family or friends [56], which are secondary control strategies. In fact, studies comparing American and Asian populations have shown that secondary control is more important than primary control in the daily lives of Asian populations [57, 58]. This could mean that the results found here on the role of (primary) control in the experience of life events in the pandemic are not generalizable to collectivistic cultures and that other factors (such as family availability) may play a more important role.

Despite these limitations, this study demonstrates the important role of perceived control in dealing with life events during exceptional times such as the pandemic. In addition, the data suggests that people of all ages are similarly affected by the pandemic when they experience significant life events, as age explained little variance when included as a predictor. A more important predictor was the control that people experienced over their life events. Thus, the context of peoples' lives might be more important for their experience in the pandemic than their age. This is in line with previous and recent suggestions to pay more attention to the context of people's lives when investigating age differences in the experience of significant life events [59].

## Supporting information

**S1 File.**
(DOCX)

## Acknowledgments

The authors thank all the participants in the study. The study was not preregistered.

## Author Contributions

**Conceptualization:** Sonja Radjenovic.

**Data curation:** Jana Nikitin.

**Formal analysis:** Sonja Radjenovic.

**Funding acquisition:** Jana Nikitin.

**Investigation:** Sonja Radjenovic.

**Methodology:** Sonja Radjenovic, Christina Ristl, Jana Nikitin.

**Supervision:** Jana Nikitin.

**Visualization:** Sonja Radjenovic.

**Writing – original draft:** Sonja Radjenovic.

**Writing – review & editing:** Christina Ristl, Jana Nikitin.

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
