## [Decision Letter · Decision Letter 0]

24 May 2022

PONE-D-22-01404Experiencing a Significant Life Event During the COVID-19 Pandemic: The Role of Perceived ControlPLOS ONE

Dear Dr. Radjenovic,

Thank you for submitting your manuscript to PLOS ONE. After careful consideration, we feel that it has merit but does not fully meet PLOS ONE’s publication criteria as it currently stands. Therefore, we invite you to submit a revised version of the manuscript that addresses the points raised during the review process. While the reviewers and I agree that the topic of the manuscript is interesting and important, there are several major concerns that need to be addressed. Please make sure to address all of the major and minor concerns outlined by each reviewer.

We look forward to receiving your revised manuscript.

Kind regards,

Neha John-Henderson

Academic Editor

PLOS ONE

2. Peer review at PLOS ONE is not double-blinded (https://journals.plos.org/plosone/s/editorial-and-peer-review-process). For this reason, authors should include in the revised manuscript all the information removed for blind review.

Reviewers' comments:

Reviewer's Responses to Questions

**Comments to the Author**

1. Is the manuscript technically sound, and do the data support the conclusions?

Reviewer #1: No

Reviewer #2: Partly

2. Has the statistical analysis been performed appropriately and rigorously? 

Reviewer #1: No

Reviewer #2: Yes

3. Have the authors made all data underlying the findings in their manuscript fully available?

Reviewer #1: No

Reviewer #2: Yes

4. Is the manuscript presented in an intelligible fashion and written in standard English?

Reviewer #1: No

Reviewer #2: Yes

5. Review Comments to the Author

Reviewer #1: Thank you for your very interesting study. However, I recommend studying basic academic writing and research methodology before resubmission.

In Introduction, please systematically summarize what is known and what is not known about perceived control in COVID-19 pandemic.

It is unclear if you assume perceived control as a predictor or outcome or both (?).

It is better to draw a conceptual framework, based on theories.

Is description written in L117-128 from your previous study or this study? If it is from this study, please include it in Results.

Gender and educations are important factors of perceived control. If you omit these variables, I’m afraid you miss important information. You have to report demographics, including gender and education. Please be based on theories, not just exploratory analyses.

L130

Please define age-heterogenous sample and how you accessed them.

Please explain more about online recruitment service. Did you use email or some social media? It would not be easy to approach older people, and I wonder how you did.

L139-145

I think this part should also be written in Results.

L146- Measures

Please describe more about scales, including reliability and validity with references if applicable.

P173-

Explanation of Tables should be included in Results.

In Method, you need to explain Analysis and Ethical consideration. Have you obtained permission from Institutional Review Board? How did you explain and obtained agreement from participants?

What was the significant level you set? There are many analyses. It is better to set lower significant level to decrease alpha error.

Did you use linear regression? Have you checked the distribution (normality) of outcome variables?

L182-195

This part should be written in Methods.

L193-195

You also have to consider alpha error. Please also exactly report the effect size, analysis, one tail or two tail, which you set in G*Power.

Please indicate number of participants in Table 1 and 2.

In Table 2, total number is 667, which is smaller than 882. If 215 participants answered others, it should be included in the Table.

In Result, we usually do not cite references. You do not need summary of results and conclusion in Results. It should be written in Discussion and Conclusion.

The contents and form of Table 3 and 4 seem also to be unusual. It is unusual to report only t, βand p in this form.

Figure 1 is unclear and hard to see.

L297-8

“The findings of the present study demonstrate that the pandemic affects how we experience significant life events.”

It is hard to say that your study indicated that pandemic affects experience of life-events since you did not compare to the situation without pandemic.

Reviewer #2: In my review, I have made several comments on measures or outcomes that are mentioned in the introduction to be related to the author's results (e.g., mental health, stress) but not utilized in analyses.

The authors utilized appropriate statistical models for their hypotheses, and included exploratory analyses that further strengthen their overall hypotheses.

All data was made available by the authors.

I have made comments on grammatical errors throughout the manuscript. Overall, the paper is written in standard English in an intelligible fashion.

6. PLOS authors have the option to publish the peer review history of their article (what does this mean?). If published, this will include your full peer review and any attached files.

Reviewer #1: No

Reviewer #2: **Yes: **Cory Counts

---

## [Author Response · Author response to Decision Letter 0]

26 Jul 2022

Please see the attached file "Response to reviewers" for all our responses to editor's and reviewers' comments.

---

## [Decision Letter · Decision Letter 1]

7 Oct 2022

PONE-D-22-01404R1Experiencing a significant life event during the COVID-19 pandemic: The role of perceived controlPLOS ONE

Dear Dr. Radjenovic,

Thank you for submitting your manuscript to PLOS ONE. After careful consideration, we feel that it has merit but does not fully meet PLOS ONE’s publication criteria as it currently stands. Therefore, we invite you to submit a revised version of the manuscript that addresses the points raised during the review process.

We look forward to receiving your revised manuscript.

Kind regards,

Michio Murakami

Academic Editor

PLOS ONE

Reviewers' comments:

Reviewer's Responses to Questions

**Comments to the Author**

1. If the authors have adequately addressed your comments raised in a previous round of review and you feel that this manuscript is now acceptable for publication, you may indicate that here to bypass the “Comments to the Author” section, enter your conflict of interest statement in the “Confidential to Editor” section, and submit your "Accept" recommendation.

Reviewer #3: (No Response)

Reviewer #4: (No Response)

2. Is the manuscript technically sound, and do the data support the conclusions?

Reviewer #3: Partly

Reviewer #4: Yes

3. Has the statistical analysis been performed appropriately and rigorously? 

Reviewer #3: I Don't Know

Reviewer #4: Yes

4. Have the authors made all data underlying the findings in their manuscript fully available?

Reviewer #3: Yes

Reviewer #4: Yes

5. Is the manuscript presented in an intelligible fashion and written in standard English?

Reviewer #3: Yes

Reviewer #4: Yes

6. Review Comments to the Author

Reviewer #3: 1.General comments

This study is a cross-sectional study of an age heterogeneous sample recruited through an online campaign in German-speaking countries. The focus of the study is very interesting, as the study reveals the hypothesis testing that control perception is an important factor in whether the COVID-19 pandemic affects the experience of important life events. However, the methodology for the purpose of this study is deficient and the research framework needs to be modified.

2.Specific comments

a)Maior

1. Although this study purports to examine whether the pandemic affected life events, there is no comparison with data from non-pandemic periods, so the pandemic is only a

conditions only. In other words, at this point, the framework of the hypothesis may be that control cognition is a useful factor that influences the experience of life events during a pandemic period.

２. Many scales are used, including outcomes, but none are standardized and there is insufficient explanation and citation of self-development scales. Many of the scales are 7 scales, but please explain in detail how cutoffs are established and aggregated. Reliability can be confirmed, but there is no validity verification and no standard of scientific evidence is apparent.

b）Minor

１．Since it is difficult to understand the attributes of the subjects of the analysis, the authors should add a table of basic attributes, socioeconomic indicators, and other variables. Also, you would do well to add a diagram of the conceptual framework of this study to make it easier to understand.

2. You exclude gender and education from your analysis because these factors do not affect your results, but life events such as changing jobs or retiring may affect them, so do you need to exclude them from your analysis?

3. You should describe in the method the contents of lines 114 through 124. Similarly, lines 124 through 125 are the contents of the result.

4. You should describe the characteristics of the elderly who participate through the online campaign, since you explore the possibility of age effects in the introduction and specifically detail the negative impact of the pandemic on the elderly. Also, in line 94, you state that you would like to shift the focus from age differences to people's life circumstances, but it would be better to state that you are focusing on both age and life circumstances.

5. In line 326, you state "older ages was associated with lower perceived contrtol over the life events", but I am not sure which result you are referring to as the basis for this statement.

６. For the results of the hierarchical multiple regression analysis in Table 3/Table 4, you should preferably not discuss them collectively as life events, but also discuss the results for each characteristic event.

Reviewer #4: I’ve read the present study with great interest in the timely and important topic during the pandemic. The authors have addressed most of the concerns of the reviewers, but I would like to raise a few points that I believe the manuscript could improve further.

INTRODUCTION

Lines 51-63: I recommend adding a bit of the COVID-19 epidemic situation or the national response to the pandemic in the study area. Because the study focuses on the COVID-19 influence and the perceived control, the social background would help to understand the study background.

Lines 105-113: Please state the aim of the study.

Lines 114-148: I think this paragraph should be written in the Methods.

METHODS

Lines 147-148: Did the authors use the term ‘subsample’ here for the present study sample (N= 882)? There is the other ‘subsample’ in line 200 and the legend of Table 2. I think the former two ‘subsamples’, who are the participants of this study, excluding those who had out-of-period events from the total of 6,688 subjects, may not be considered a subsample. In the present study, the number of the participants for overall analysis was 882, and for the analysis of seven clusters of life events (a subsample?) was 667.

Line 175: I cannot understand ‘the six items.’

Data-Analytical Strategy,

Please clarify the regression model. It would be easier to understand the results section if you explained the age-adjusted model and how to evaluate its impact. Please explain how the age and influence intensity were treated as control variables (e.g. per decade or a continuous variable).

Please explain how the authors did exploratory analyses. For this point, the paragraph, Lines 287-292, might be available and should be placed in the Data-Analytical Strategy.

RESULTS

The authors did not control for gender. To strengthen the reason for that, please explain the specific index (i.e. not ‘no systematic changes’ but not changes in regression coefficients or delta R2). I also recommend including gender in the Descriptive statistics (Supplementary Table 2).

Table 3: Are the βs for the positive/negative influence and perceived control the regression coefficients in the age and overall influence intensity-adjusted model or in the overall influence intensity-adjusted model (not adjusted by age)? Please clarify.

DISCUSSION

The authors could discuss the relationship between the study results and cultural or social characteristics in the study area (Europe) to help understanding the mechanism of the results.

7. PLOS authors have the option to publish the peer review history of their article (what does this mean?). If published, this will include your full peer review and any attached files.

Reviewer #3: No

Reviewer #4: No

---

## [Author Response · Author response to Decision Letter 1]

21 Nov 2022

Please see the "Response to reviewers" doc file uploaded. Thank you.

---

## [Decision Letter · Decision Letter 2]

12 Dec 2022

PONE-D-22-01404R2Experiencing a significant life event during the COVID-19 pandemic: The role of perceived controlPLOS ONE

Dear Dr. Radjenovic,

Thank you for submitting your manuscript to PLOS ONE. After careful consideration, we feel that it has merit but does not fully meet PLOS ONE’s publication criteria as it currently stands. Therefore, we invite you to submit a revised version of the manuscript that addresses the points raised during the review process.

Please see Additional Editor Comments.

We look forward to receiving your revised manuscript.

Kind regards,

Michio Murakami

Academic Editor

PLOS ONE

Journal Requirements:

Additional Editor Comments:

1. Figure 2 is mentioned in the text before Figure 1. Please renumber the figures.

2. For Figure 1, each plot probably represents more than one participant, because of the same value among the participants. Please use a bubble plot to show the number of participants.

3. There is an insufficient description to the limitations of the bias due to older people participating in the online campaign (older people in this study are more likely to have characteristics that make them more familiar with online surveys than older people in general).

Reviewers' comments:

Reviewer's Responses to Questions

**Comments to the Author**

1. If the authors have adequately addressed your comments raised in a previous round of review and you feel that this manuscript is now acceptable for publication, you may indicate that here to bypass the “Comments to the Author” section, enter your conflict of interest statement in the “Confidential to Editor” section, and submit your "Accept" recommendation.

Reviewer #4: All comments have been addressed

2. Is the manuscript technically sound, and do the data support the conclusions?

Reviewer #4: Yes

3. Has the statistical analysis been performed appropriately and rigorously? 

Reviewer #4: I Don't Know

4. Have the authors made all data underlying the findings in their manuscript fully available?

Reviewer #4: Yes

5. Is the manuscript presented in an intelligible fashion and written in standard English?

Reviewer #4: Yes

6. Review Comments to the Author

Reviewer #4: (No Response)

7. PLOS authors have the option to publish the peer review history of their article (what does this mean?). If published, this will include your full peer review and any attached files.

Reviewer #4: No

---

## [Author Response · Author response to Decision Letter 2]

14 Dec 2022

Please see the attached doc file "Response to reviewers".

---

## [Editor Report · Decision Letter 3]

16 Dec 2022

Experiencing a significant life event during the COVID-19 pandemic: The role of perceived control

PONE-D-22-01404R3

Dear Dr. Radjenovic,

We’re pleased to inform you that your manuscript has been judged scientifically suitable for publication and will be formally accepted for publication once it meets all outstanding technical requirements.

Kind regards,

Michio Murakami

Academic Editor

PLOS ONE
---

## [Editor Report · Acceptance letter]

23 Dec 2022

PONE-D-22-01404R3 

Experiencing a significant life event during the COVID-19 pandemic: The role of perceived control 

Dear Dr. Radjenovic:

I'm pleased to inform you that your manuscript has been deemed suitable for publication in PLOS ONE. Congratulations! Your manuscript is now with our production department. 

Kind regards, 

on behalf of

Dr. Michio Murakami 

Academic Editor

PLOS ONE